# Optimal power–heat–carbon scheduling strategy for interconnected heterogeneous multi-microgrid considering hydrogen fuel cell vehicles

**Dahu Li[1,2], Zirui Shao [1]\*, Wentao Huang[1], Bohan Zhang[1], Jun He[1], Xinyu Liu[1]**

**1** Hubei Collaborative Innovation Center for High-Efficiency Utilization of Solar Energy, Hubei University of Technology, Wuhan, China, **2** State Grid Hubei Electric Power Co, Ltd, Wuhan, China

\* 102210320@hbut.edu.cn

**Data Availability Statement:** All relevant data are within the manuscript and its Supporting information files.

## Abstract

The scale of multi-microgrid (MMG) and hydrogen fuel cell vehicles (HFCVs) is increasing dramatically with the increase in the new energy penetration ratio, and developing an integrated energy system containing a multi-microgrid for hydrogen fuel vehicles brings great challenges to power grid operation. Focusing on the difficulties of the access of multiple microgrids for the low-carbon and economic operation of the system, this paper proposes an optimal interconnected heterogeneous multi-microgrid power–heat–carbon scheduling strategy for hydrogen-fueled vehicles. Firstly, an HFCV model is established, and then an optimal scheduling model is constructed for the cooperative trading of power–heat–carbon in a multi-microgrid, on the basis of which the low-carbon economic operation of the multi-microgrid is realized. The results of the case study show that the scheduling strategy in this paper reduces carbon emissions by about 7.12% and costs by about 3.41% compared with the independent operation of the multi-microgrid. The degrees of interaction of each multi-microgrid are also analyzed under different HFCV penetration rates.

## 1. Introduction

In order to achieve the goal of "double carbon", integrated energy with the characteristics of multi-energy complementarity, energy class utilization, etc., provides an important means to achieve low-carbon transition [1]. As an important manifestation of integrated energy systems, the multi-microgrid can realize the complementary use and optimal matching of multiple energy sources [2]. The multi-microgrid plays an important role in enhancing the utilization of new energy sources in the system and reducing carbon emissions through energy sharing among microgrids [3].

There is some research on cooperative operation within multi-microgrids. Ref. [4] evaluated various energy scheduling and control strategies for multi-microgrids in terms of interactive energy trading, multi-energy management and resilient operations. Ref. [5] constructed a distributed energy trading management strategy for multi-microgrids based

**Funding:** The author(s) received no specific funding for this work.

**Competing interests:** The authors have declared that no competing interests exist.

on the ADMM algorithm. Ref. [6] proposed a cooperative game method for coordinating the operation of multi-microgrids. Ref. [7] considered a multi-microgrid optimization model for distributed robust optimization to solve the uncertainty optimization problem for renewable energy, loads and electric vehicles. Ref. [8] proposed a two-stage trading model for microgrid clusters based on the price trading mechanism and conditional value-at-risk (CVaR) theory, introducing the CVaR theory to quantify the potential risk of trading defaults and reduce the operating costs of microgrid clusters. Ref. [9] proposed a new integrated concept of risk-seeking/-averse preferences and established an unexpected profit-aware stochastic dispatch model for industrial virtual power plants (IVPP) based on this type of risk preference, which enables the joint management of potentially high profits and extreme losses.

The above literature does not consider the impact of specific microenergy networks on the power system. With the diversified development of cities, significant differences emerge in the load composition and other aspects of multi-microgrids with different service objects, such as residential areas, commercial areas, industrial areas, etc., and the differences in equipment coupling and energy conversion characteristics between different types of multi-microgrid raise the possibility of optimizing and adjusting the spatial–temporal complementarity of energies between multi-microgrid systems [10].

Ref. [11] considered an optimal multi-temporal- and -spatial-scale scheduling method for integrated energy containing industrial, commercial, and residential microenergy grids, and Ref. [12] investigated the optimal operation mode for different functional distribution grids in Southwest China. Ref. [13] constructed models for different functional areas and proposed a non-zero-sum game-based optimal scheduling strategy for power–heat–carbon in interconnected heterogeneous multi-microgrid systems.

With the continuous development of "green hydrogen", the hydrogen energy industry is now not only used in the electric power system, but also in the transportation field [14]. HFCVs are developing rapidly due to the high specific energy of hydrogen, which can be used to travel long distances, and its short charging time, which is often less than 5 minutes [15]. Ref. [16] constructed a dynamic model of an electrolyzer for hydrogen fuel vehicles and hydrogen loading. Ref. [17] investigated the modeling of rooftop PV–hydrogen-fueled vehicles to achieve the goal of net-zero-energy residential buildings. Ref. [18] studied the use of off-grid solar charging stations for electric and hydrogen-fueled vehicles to meet the demand for hydrogen-fueled vehicles in the region. With the continuous development of hydrogen-fueled vehicles, hydrogen-powered heavy trucks have started to appear in industrial parks [19]. Table 1 compares the advantages of the proposed approach with the findings of the existing literature on multi-microenergy grid systems.

Based on the above research, this paper considers the microgrid functional area's structure and load differences, and at the same time introduces hydrogen truck loads into the industrial multi-microgrid, formulating an optimal scheduling strategy for the multi-microgrid system that includes power, heat, and carbon energy trading. The article firstly constructs a hydrogen fueling demand model for hydrogen fuel trucks, then constructs an optimal scheduling model for power–heat–carbon co-transactions in multi-microgrids, before finally solving the model and proving the effectiveness of the proposed strategy through arithmetic examples. Compared with the existing literature, this paper has conducted further research on the following aspects:

1. Considering the characteristic endowments of residential, commercial and industrial areas, we construct specific power and heat models for each functional area, and design the

**Table 1. Comparison of the proposed methodology with the existing literature on multi-microenergy networks.**

| Ref | MMG | Functional areas | HFCV | Energy type | | |
|---|---|---|---|---|---|---|
| | | | | Power | Heat | Hydrogen |
| [5] | √ | × | × | √ | × | × |
| [6] | √ | × | × | √ | × | × |
| [7] | √ | × | × | × | × | × |
| [8] | √ | × | × | × | × | × |
| [9] | √ | √ | × | √ | × | × |
| [11] | √ | √ | × | √ | √ | √ |
| [12] | √ | × | × | √ | × | × |
| [13] | √ | √ | × | √ | √ | × |
| [17] | × | × | √ | √ | √ | √ |
| [18] | × | × | √ | √ | × | √ |
| Proposed method | √ | √ | √ | √ | √ | √ |

operation and scheduling framework of a heterogeneous multi-microgrid system. We also construct distributed photovoltaic power generation models for residential and commercial areas, and wind power and photovoltaic power generation models for industrial areas, so as to give full play to the unique geographic advantages and characteristics of each functional area.

2. Considering the energy complementarity characteristics of heterogeneous functional zones, we formulate an electricity–heat–carbon trading strategy for the multi-microgrid system, making full use of the differences between functional areas, and promote trading between functional zones to comprehensively reduce the operating costs and carbon emissions of the system.

3. We construct a model for hydrogen energy loads containing hydrogen fuel vehicles and industrial areas, so as to achieve the diversified use of hydrogen energy through the consumption of renewable energy and the development of green industries.

## 2. Framework for coordinated operation of multi-microgrid systems

In this paper, a multi-microgrid system is modeled by interconnecting residential, commercial and industrial areas. It is shown in Fig 1.

Based on the similarity between residential and commercial loads, this paper constructs a topology model for a multi-microgrid in residential and commercial areas, utilizing equipment such as combined heat and power (CHP), photovoltaic (PV) units, gas boilers (GB), batteries (BAT), and thermal storage equipment. Unlike residential and commercial areas, industrial areas are usually located in open areas, in which wind power (WT) can be utilized more efficiently, and in addition, here, cogeneration units and gas boilers are set up as the main electrical and thermal energy supply equipment. Due to the emergence of hydrogen loads and hydrogen heavy trucks in industrial parks, electric hydrogen production units (P2H) are now being accessed for hydrogen production.

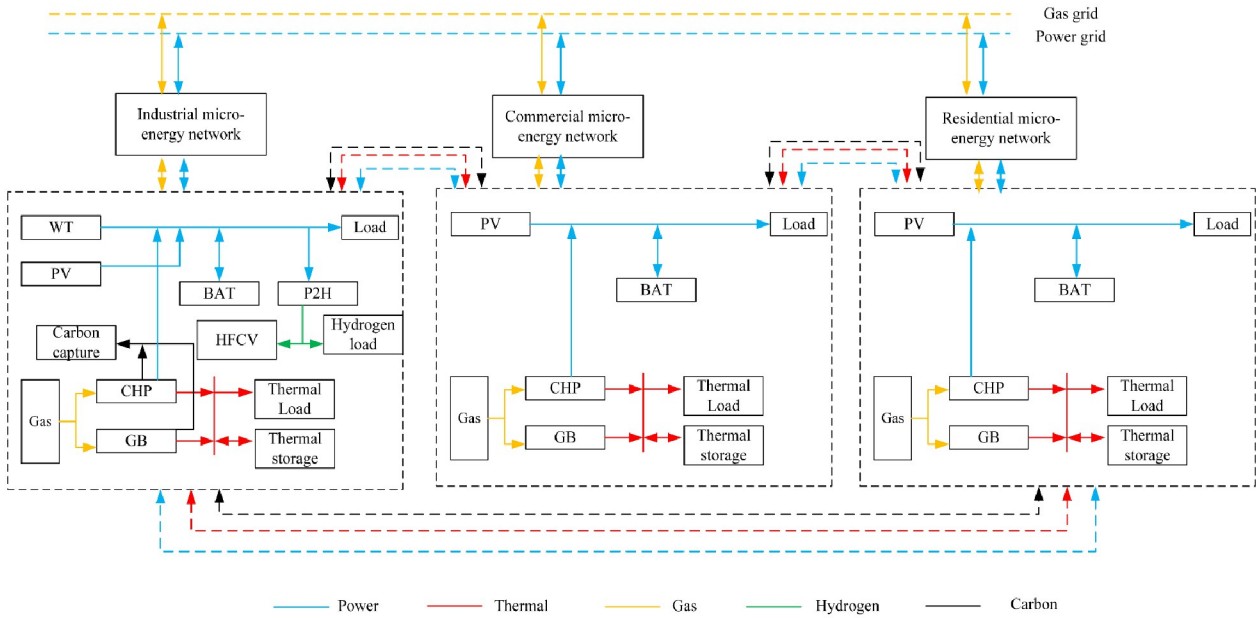

**Fig 1. Block diagram of multi-microgrid structure.**

## 3. Hydrogen fueling demand model for hydrogen-fueled vehicles

For W hydrogen heavy trucks, the moment of hydrogen refueling is defined as follows [20]:

$$\begin{cases} T_z = \sum_{w=1}^{W} T_w \\ T_w = \dfrac{S_{jq,w} - S_{cs,w}}{S_{d,w}} \end{cases} \tag{1}$$

In Eq (1), $T_z$ is the total set of hydrogen refueling moments; $T_w$ is the hydrogen refueling moment of the wth hydrogen truck; $S_{jq,w}$ is the hydrogen refueling mileage of the wth hydrogen truck; $S_{cs,w}$ is the initial mileage of the wth hydrogen truck on the same day; and $S_{d,w}$ is the mileage per unit of the wth hydrogen truck. The total number of vehicles requiring hydrogen refueling in a day is defined as follows:

$$N_z = \sum_{w=1}^{w} n_w \tag{2}$$

$$n_w = \begin{cases} 1 & , 0 \le T_w \le 24 \\ 0 & , T_w > 24 \end{cases} \tag{3}$$

In Eq (2), $N_z$ is the total number of hydrogen vehicles refueling in a day; $n_w$ is the hydrogen refueling judgment coefficient of the wth hydrogen heavy truck—if the moments are distributed within 24 hours, the hydrogen refueling condition is satisfied, and otherwise, no hydrogen refueling is performed.

The total amount of hydrogenation $M_q$ in a day is defined as follows:

$$M_q = \sum_{w=1}^{W} \frac{n_w S_{jq,w} m_w}{S_{z,w}} \tag{4}$$

In Eq (3), $m_w$ is the hydrogen storage capacity of the wth hydrogen heavy truck; $S_{z,w}$ is the mileage of the wth hydrogen heavy truck under full hydrogen condition.

# 4. Optimal scheduling model for power–heat–carbon co-trading in a multi-microgrid

## 4.1. Multi-microgrid power–heat–carbon synergistic trading mechanism

Multi-microgrid trading in this paper comprises the three elements of power, heat and carbon. Among these, there are two modes of power and heat energy trading: the distributed trading of energy, and energy trading with higher-level energy networks. In the process of multi-microgrid transactions, the power transaction $P_{ij}(t)$ between microenergy network $i$ and microenergy network $j$ can be expressed as follows:

$$\begin{cases} P_{ij}(t) \in \left\{ \theta_1 P_{ij}^{buy}(t), \theta_2 P_{ij}^{sell}(t) \right\} \\ 0 \leq \theta_1 + \theta_2 \leq 1 \end{cases} \tag{5}$$

In Eq (5), we see the two behaviors of purchasing and selling power for microenergy network $i$ and micro-grid $j$. When $P_{ij}^{buy}(t)$, microenergy network $i$ buys power from microenergy network $j$, and $P_{ij}(t)$ is positive; when $P_{ij}^{buy}(t)$, microenergy network $i$ sells power to microenergy network $j$, and $P_{ij}(t)$ is negative; $\theta_1, \theta_2$ is a binary 0–1 variable. Thermal energy trading satisfies the same law.

In order to avoid the waste and insufficiency of carbon allowances, a carbon allowance trading mechanism is set to realize equilibrium in the carbon trading market. After the scheduling strategy for each time period of the microenergy network has been determined, the carbon quota determined by the system of power generation often deviates from the actual carbon emissions. According to the carbon quota benchmark, the part of the microenergy network that emits carbon in excess and the part that is below the quota can be traded for carbon emission rights at a certain price, which improves the overall economy at the same time as the green credentials:

$$\beta_{c,i} = \begin{cases} \psi E_{i,w}, -l \leqslant E_{i,w} \leqslant l \\ \psi(1+\mu)(E_{i,w} - l) + \psi l, l < E_{i,w} \leqslant 2l \\ \psi(1+2\mu)(E_{i,w} - 2l) + \psi(2+\mu)l, 2l < E_{i,w} \leqslant 3l \\ \psi(1+3\mu)(E_{i,w} - 3l) + \psi(3+3\mu)l, 3l < E_{i,w} \leqslant 4l \\ \psi(1+4\mu)(E_{i,w} - 4l) + \psi(4+6\mu)l, E_{i,w} > 4l \end{cases} \tag{6}$$

In Eq (6), $\beta_{c,i}$ is the stepped carbon trading cost of each microenergy network; $\psi$ is the carbon trading base price; $\mu$ is the price growth rate; $l$ is the length of the interval per unit of carbon emissions.

## 4.2. Objective function

The scheduling objective of the multi-microgrid system is to reduce the operating cost and environmental pollution of the system while satisfying all types of loads as much as possible.

Taking 1 h as the scheduling scale, the objective function is to minimize the operating cost $f^u$ of the whole multi-microgrid system, which includes the cost of purchasing energy $f^b$, the cost of microenergy grid synergy transaction $f^m$, and the cost of carbon emission right transaction $f^c$, which can be obtained as follows:

$$min f^u = f^b + f^m + f^c \tag{7}$$

This paper defaults to microgrid 1 for residential areas, microgrid 2 for commercial areas, and microgrid 3 for industrial areas.

### 4.2.1. Cost of energy purchases.

$$f^b = \sum_{i=1}^{3} \sum_{t=1}^{T} \left( C_e(t) P_{b,i}(t) + C_g(t) G_{b,i}(t) \right) \tag{8}$$

In Eq (8), $C_e(t)$ and $C_g(t)$ are the purchased power and gas prices at moment t; $P_{b,i}(t)$ and $G_{b,i}(t)$ are the purchased power and gas power of the ith microenergy network at moment t, respectively.

### 4.2.2. Microenergy network synergy transaction costs.

$$f^m = \sum_{i=1}^{3} \sum_{t=1}^{T} \left( r_e(t)(P_{ij}^{buy}(t) - P_{ij}^{sell}(t)) + r_h(t)(H_{ij}^{buy}(t) - H_{ij}^{sell}(t)) \right) \tag{9}$$

In Eq (9), $r_e(t)$ and $r_h(t)$ are the interactive prices of electrical and thermal energy of the microenergy network at time t, respectively.

### 4.2.3. Carbon emissions trading costs.

The carbon emissions of each microenergy network mainly come from the carbon emissions to the upper grid, CHP and GB.

The actual carbon emissions of each microenergy network can be expressed as follows:

$$\begin{cases} E_i^* = \sum_{t=1}^{T} \left( E_{b,i}^*(t) + E_{c,i}^*(t) \right) \\ E_{b,i}^*(t) = \delta_e P_{b,i}(t) \\ E_{c,i}^*(t) = \delta_g (G_{chp,i}(t) + G_{gb,i}(t)) \end{cases} \quad i = 1, 2, 3 \tag{10}$$

In Eq (10), $E_i^*$ is the actual carbon emission of each microenergy network; $E_{b,i}^*(t)$ is the carbon emission related to purchasing power from the grid at moment t; $E_{c,i}^*(t)$ is the carbon emission of CHP and GB at moment t; $\delta_e$ is the carbon emission coefficient of purchasing power from the grid; $\delta_g$ is the carbon emission coefficient of natural gas; and $G_{chp,i}(t)$ and $G_{gb,i}(t)$ are the natural gas input power of CHP and GB, respectively.

The carbon quota model for each microenergy network can be expressed as follows:

$$\begin{cases} E_i = \sum_{t=1}^{T} \left( E_{b,i}(t) + E_{c,i}(t) \right) \\ E_{b,i}(t) = \alpha_e P_{b,i}(t) \\ E_{c,i}(t) = \alpha_g (G_{chp,i}(t) + G_{gb,i}(t)) \end{cases} \quad i = 1, 2, 3 \tag{11}$$

In Eq (11), $E_i$ is the carbon allowance of each microenergy network; $E_{b,i}(t)$ is the carbon allowance brought by purchasing power from the grid at moment t; $E_{c,i}(t)$ is the carbon emission of CHP and GB at moment t; $\alpha_e$ is the carbon allowance coefficient of purchasing power from the grid; and $\alpha_g$ is the carbon allowance coefficient of natural gas.

The amount of carbon emission rights traded of each microenergy network can be expressed as follows:

$$E_{i,w} = E_i^* - E_i \quad i = 1, 2, 3 \tag{12}$$

In Eq (12), $E_{i,w}$ is the amount of carbon emission rights traded by each microenergy network.

Carbon credit transaction costs across microgrids.

$$f_c = \sum_{i=1}^{3} E_{i,w} \tag{13}$$

## 4.3. Restrictive condition

### 4.3.1. CHP constraints.
The CHP unit consists mainly of a gas turbine and boiler that provides power and heat to the outside, and is mathematically modeled as follows [21]:

$$\begin{cases} H_{chp,i}(t) = \eta_{chp}^{h,e} P_{chp,i}(t) \\ G_{chp,i}(t) = \dfrac{1}{H_{ng}} \dfrac{P_{chp,i}(t)}{\beta_{chp}} \qquad i = 1, 2, 3 \\ 0 \leq H_{chp,i}(t) \leq H_{chp,i}^{\max} \end{cases} \tag{14}$$

In Eq (14), $H_{chp,i}(t)$ is the thermal power output by the ith microgrid CHP unit; $\eta_{chp}^{h,e}$ is the thermoelectric conversion efficiency of the CHP unit; $P_{chp,i}(t)$ is the power generation of the ith microgrid CHP unit; $G_{chp,i}(t)$ is the amount of natural gas consumed by the ith microgrid CHP unit; $\beta_{chp}$ is the power generation efficiency of the CHP unit, and $H_{chp,i}^{max}$ is the maximum value of heat production of the ith microenergy network CHP unit.

### 4.3.2. GB constraints.
The output of GB is modeled as follows [22]:

$$\begin{cases} H_{gb,i}(t) = \eta_{gb} G_{gb,i}(t) \qquad i = 1, 2, 3 \\ 0 \leq H_{gb,i}(t) \leq H_{gb,i}^{\max} \end{cases} \tag{15}$$

In Eq (15), $H_{gb,i}(t)$ is the thermal power output by the GB unit of the ith microenergy network; $\eta_{gb}$ is the thermal efficiency of the GB, and $H_{gb,i}^{max}$ is the maximum value of the power of the GB of the ith microenergy network.

### 4.3.3. Electrolytic tank constraints.

$$\begin{cases} R_{h,i}(t) = \eta \, P_{el,i}(t) \qquad i = 1, 2, 3 \\ 0 \leq P_{el,i}(t)) \leq P_{el,i}^{\max} \end{cases} \tag{16}$$

In Eq (16), $P_{el,i}(t)$ is the input power of the ith microenergy grid electrolyzer; $R_{h,i}(t)$ is the hydrogen production power of the ith microenergy grid; and $P_{el,i}^{max}$ is the maximum value of the input power of the ith microenergy grid electrolyzer.

**4.3.4. Energy storage constraints.** Energy storage devices play an important role in energy management in multi-microgrids, which are generally modeled as follows [23]:

$$E_{x,i}(t+1) = \begin{cases} E_{x,i}(t)(1-\delta_x) + P_{x,i,c}(t)\eta_{x,c} \\ E_{x,i}(t)(1-\delta_x) - P_{x,i,d}(t)/\eta_{x,d} \end{cases} \quad x \subseteq \{e,h\} \tag{17}$$

$$\begin{cases} E_{x,i,\min} \leqslant E_{x,i}(t) \leqslant E_{x,i,\max} \\ 0 \leqslant P_{x,c,i}(t) \leqslant \lambda_{x,c,i} P_{x,c,i,\max} \\ 0 \leqslant P_{x,d,i}(t) \leqslant \lambda_{x,d,i} P_{x,d,i,\max} \qquad i = 1,2,3 \\ E_{x,i}(0) = E_{x,i}(24), \\ \lambda_{x,c,i} + \lambda_{x,d,i} = 1 \end{cases} \tag{18}$$

In Eqs (17) and (18), x is the type of electric and thermal energy storage, respectively. $E_x(t)$ and $E_x(t+1)$ are the energy states before and after energy charging and discharging, respectively; $\eta_{x,c}, \eta_{x,d}$ and $\delta_x$ are the charging efficiency, discharging efficiency and energy loss coefficient, respectively; $P_{x,i,c}(t)$ and $P_{x,i,d}(t)$ are the charging and discharging power at time t, respectively; $E_{x,i,max}$ and $E_{x,i,min}$ are the upper and lower limit values of the energy storage device, respectively; $P_{x,c,i,max}$ and $P_{x,d,i,max}$ are the maximum values of charging and discharging energies, respectively; $\lambda_{x,c,i}$ and $\lambda_{x,d,i}$ are the binary variables of charging and discharging energy, respectively.

**4.3.5. Energy purchase upper and lower bound constraints.**

$$\begin{cases} 0 \leqslant P_{b,i}(t) \leqslant P_{b,i}^{\max}, 0 \leqslant \sum_{i=1}^{3} P_{b,i}(t) \leqslant P_b^{\max} \\ \qquad\qquad\qquad\qquad\qquad\qquad\qquad i = 1,2,3 \\ 0 \leqslant G_{b,i}(t) \leqslant G_{b,i}^{\max}, 0 \leqslant \sum_{i=1}^{3} G_{b,i}(t) \leqslant G_b^{\max} \end{cases} \tag{19}$$

In Eq (19), $P_{b,i}^{max}$ and $G_{b,i}^{max}$ denote the maximum values of power and natural gas purchased from the outside world for the ith microenergy network, respectively; $P_b^{max}$ and $G_b^{max}$ are the maximum values of power and natural gas purchased for the whole system.

**4.3.6. Energy interaction constraints.**

$$\begin{cases} P_{ij,\min}^{buy} \leq P_{ij}^{buy}(t) \leq P_{ij,\max}^{buy} \\ P_{ij,\min}^{sell} \leq P_{ij}^{sell}(t) \leq P_{ij,\max}^{sell} \end{cases} \tag{20}$$

$$\begin{cases} H_{ij,\min}^{buy} \leq H_{ij}^{buy}(t) \leq H_{ij,\max}^{buy} \\ H_{ij,\min}^{sell} \leq H_{ij}^{sell}(t) \leq H_{ij,\max}^{sell} \end{cases} \tag{21}$$

In Eqs (20) and (21), $P_{ij,max}^{buy}$ and $P_{ij,min}^{buy}$ are the maximum and minimum values of the ith microenergy grid power interaction purchase; $P_{ij,max}^{sell}$ and $P_{ij,min}^{sell}$ are the maximum and minimum values of the ith microenergy grid power interaction sale; $H_{ij,max}^{buy}$ and $H_{ij,min}^{buy}$ are the maximum and minimum values of the ith microenergy grid heat interaction purchase; $H_{ij,max}^{sell}$ and $H_{ij,min}^{sell}$ are the maximum and minimum values of the ith microenergy grid heat interaction sale.

**4.3.7. Energy balance constraint.** The constraints on the electrical, thermal, and natural gas balances of the residential, commercial, and industrial microenergy networks are shown in Eqs (22), (23) and (24), and the hydrogen balance of the industrial microenergy network is

shown in Eq (25):

$$
\begin{cases}
P_{pv,i}(t) + P_{chp,i}(t) + P_{e,d,i}(t) + P_{ij}^{buy}(t) + P_{b,i}(t) \\
\quad = P_{load,i}(t) + P_{ij}^{sell}(t) + P_{e,c,i}(t) \quad\quad i = 1,2 \\
P_{wt,i}(t) + P_{pv,i}(t) + P_{chp,i}(t) + P_{e,d,i}(t) + P_{ij}^{buy}(t) + P_{b,i}(t) \\
\quad = P_{load,i}(t) + P_{ij}^{sell}(t) + P_{el,i}(t) + P_{e,c,i}(t) \quad\quad i = 3
\end{cases}
\tag{22}
$$

In Eq (22), $P_{pv,i}(t)$ is the photovoltaic output of the ith microenergy network; $P_{load,i}(t)$ is the electrical load of the ith microenergy network; $P_{wt,i}(t)$ is the wind power output of the ith microenergy network.

$$
\begin{aligned}
H_{gb,i}(t) + H_{chp,i}(t) + H_{h,d,i}(t) + H_{ij}^{buy}(t) \\
= H_{load,i}(t) + H_{ij}^{sell}(t) + H_{h,c,i}(t) \quad i = 1,2,3
\end{aligned}
\tag{23}
$$

In Eq (23), $H_{h,c,i}(t)$ and $H_{h,d,i}(t)$ of the ith microenergy network are the heat charging and discharging power of the heat storage device; $H_{load,i}(t)$ is the heat load of the ith microenergy network.

$$
G_{gb,i}(t) + G_{chp,i}(t) = G_{b,i}(t) \quad\quad i = 1,2,3
\tag{24}
$$

$$
R_h(t) = R_c(t) + R_{load}(t)
\tag{25}
$$

In Eqs (24) and (25), $R_c(t)$ is the hydrogen refueling power of the hydrogen-fueled vehicle; $R_{load}(t)$ is the hydrogen load.

## 4.4. Model solution

Following the above analysis, it can be seen that the objective function and constraints in this paper are linear, and none of them involve the multiplication term of binary variables and continuous variables. The commercial solver Cplex integrates the advantages of optimization algorithms such as the branch-and-bound method and the cut-plane method, and it has the ability to solve mixed-integer linear programming problems quickly. Therefore, CPLEX 12.6.0 was employed in MATLAB R2016b to solve the model.

# 5. Calculus analysis

In this paper, a simulation study is carried out on the multi-functional district microenergy network shown in Fig 1, and an arithmetic example is analyzed based on an improved real-world project, wherein the penetration rate of hydrogen-fueled vehicles is 16%. The simulation parameters are shown in Tables 2–5, and the electric heat load and price are shown in Figs 2 and 3.

## 5.1. Calculus analysis

**5.1.1. Analysis of movement control results.** The results of the optimization of residential, commercial, and industrial microgrids are shown in Figs 4–6. Residential microgrids are limited by the capacity of the CHP units and the working hours of the PVs, which result in the need to purchase power from the main grid every day to satisfy their own energy demand. Due to the principle of the CHP unit's "heat for power", the commercial microgrid undergoes heat abandonment, and after considering the interaction between microenergy networks, the commercial microgrid sends the excess heat to the industrial and residential microgrids to

**Table 2. Energy storage parameters.**

| Installations | Capacities/kW | Upper and lower capacity limits/% | Efficiency/% |
|---|---|---|---|
| Power storage equipment | 450 | 90, 10 | 95 |
| Heat storage equipment | 500 | 90, 10 | 95 |

**Table 3. Equipment parameters.**

| Parameters | Value/kW | Parameters | Value/kW |
|---|---|---|---|
| $H_{chp,i}^{max}$, $H_{gb,i}^{max}$ | 500 | $P_{ij,max}^{sell}$, $P_{ij,min}^{sell}$ | 200, 0 |
| $P_{el,i}^{max}$ | 500 | $P_{ij,max}^{buy}$, $P_{ij,min}^{buy}$ | 200, 0 |
| $P_{b,i}^{max}$, $P_b^{max}$ | 1000, 3000 | $H_{ij,max}^{buy}$, $H_{ij,min}^{buy}$ | 200, 0 |
| $G_{b,i}^{max}$, $G_b^{max}$ | 1000, 3000 | $H_{ij,max}^{sell}$, $H_{ij,min}^{sell}$ | 200, 0 |

**Table 4. Energy conversion efficiency parameters.**

| Parameters | Value |
|---|---|
| $\eta_{chp}^{h,e}$ | 0.9 |
| $\beta_{chp}$ | 0.9 |
| $\eta_{gb}$ | 0.6 |
| $\eta$ | 0.6 |
| $\eta_{x,c}$, $\eta_{x,d}$, $\delta_x$ | 0.01 |

meet their energy shortages. The industrial microgrid experiences a large energy loss resulting from power–hydrogen conversion due to the hydrogen load and the demand of hydrogen-fueled vehicles, which leads to an increase in its power load, while its power deficit is partly purchased by the main grid and partly acquired by the commercial microgrid. Residential, commercial, and industrial microgrids all store a portion of their power during periods of higher power output and lower load to cope with the lack of power output during peak load periods, thus realizing the time-shifting of power.

The interactive operation of microgrids with different attributes within the system can greatly improve the utilization rate of renewable energy, reduce the phenomena of wind and light abandonment, and realize low-carbon operation. At the same time, it reduces dependence on the higher-level grid, effectively reduces the operating pressure of the entire grid system, and improves the reliability of system operation. Power trading within the system reduces

**Table 5. Other parameters.**

| Parameters | Value |
|---|---|
| $\alpha_e$ | 0.728 kg (kW h) |
| $\alpha_g$ | 0.367 kg (kW h) |
| $\delta_e$ | 1.08 kg (kW h) |
| $\delta_h$ | 0.65 kg (kW h) |
| $l$ | 2000 kg |
| $\mu$ | 0.25 |
| $\psi$ | CNY 0.252/kg |

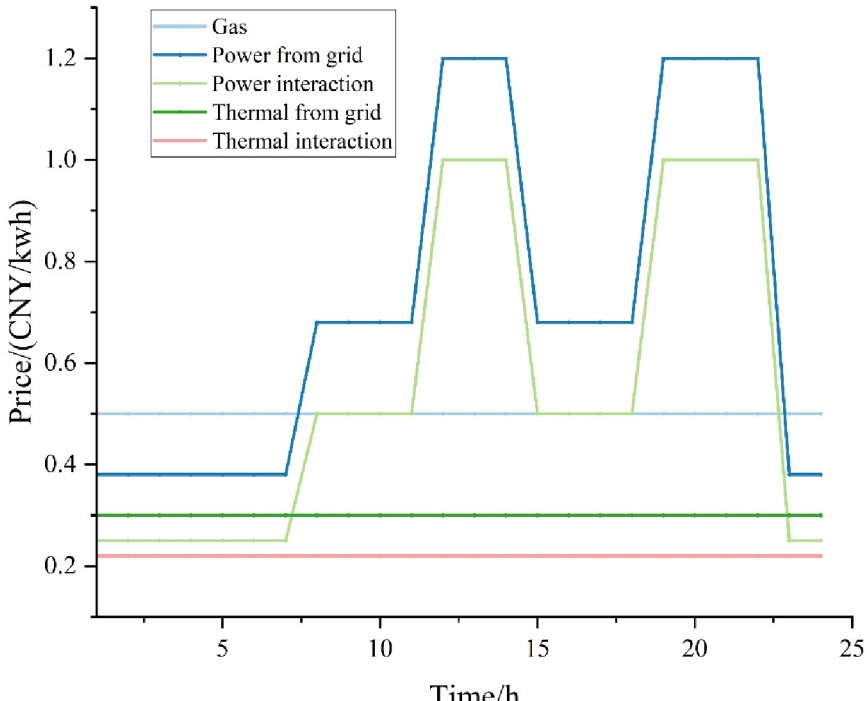

**Fig 2. Price parameters chart.**

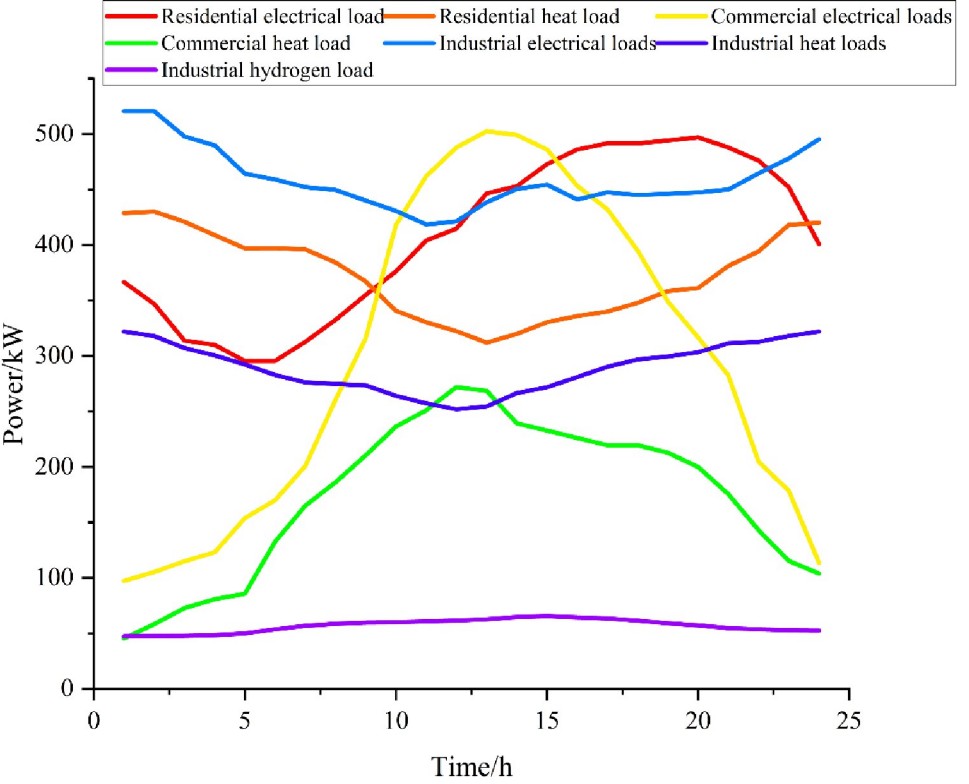

**Fig 3. Load forecast chart.**

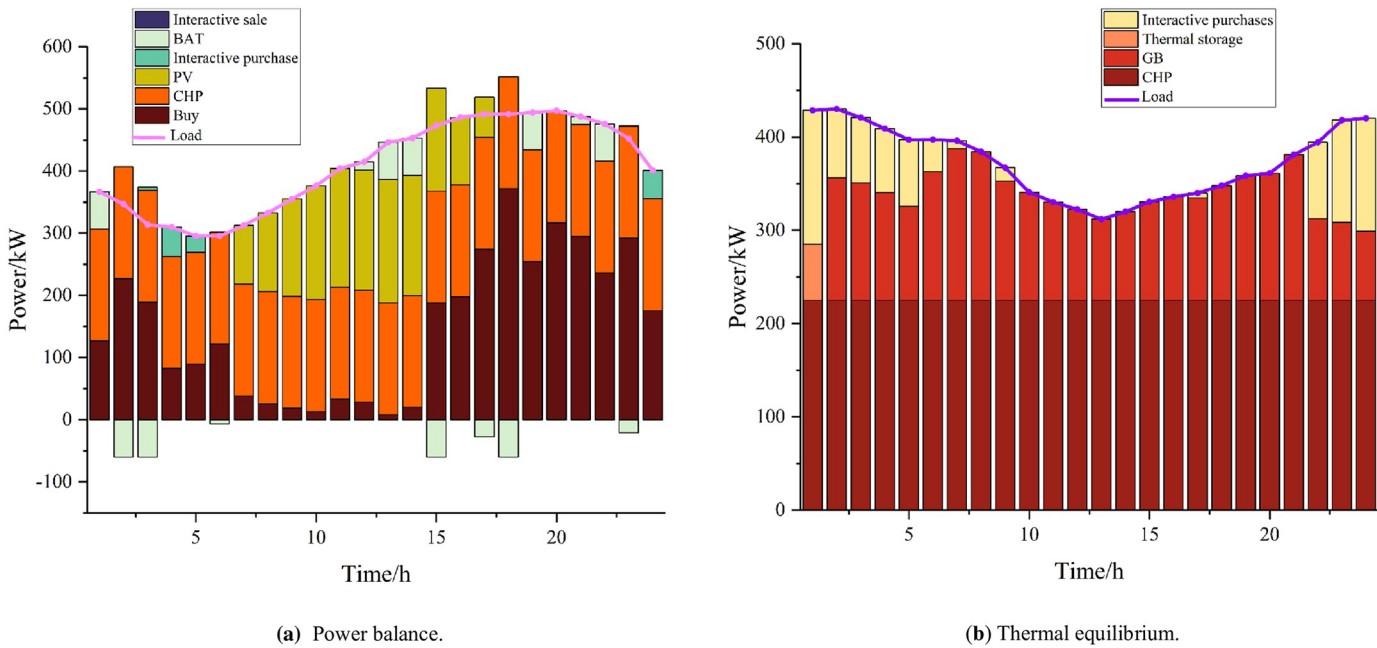

**(a)** Power balance.

**(b)** Thermal equilibrium.

**Fig 4. Resident microgrid.**

the costs of transporting power and power consumption, enhances the income from power sales, and improves the economy of the entire microgrid cluster.

Considering the synergistic nature of the overall operation of a multi-microgrid system, this paper analyzes the data via different strategies, from the perspectives of individual

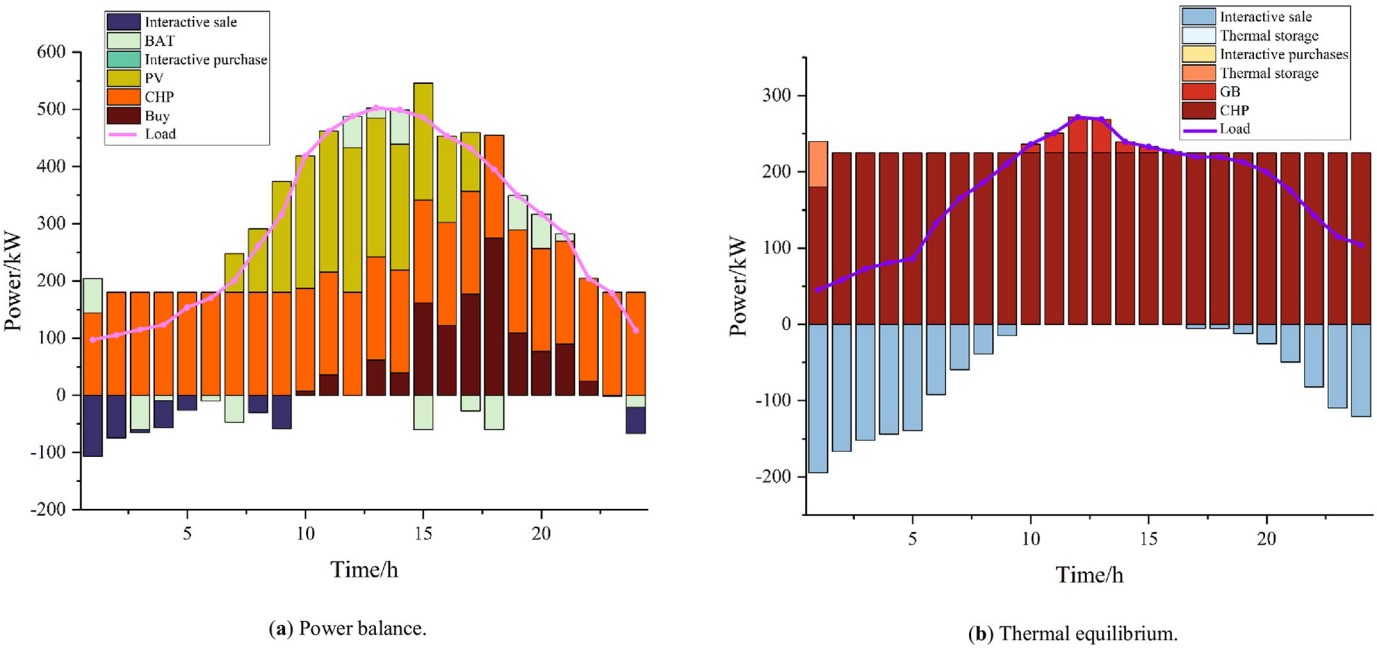

**(a)** Power balance.

**(b)** Thermal equilibrium.

**Fig 5. Business micro-networks.**

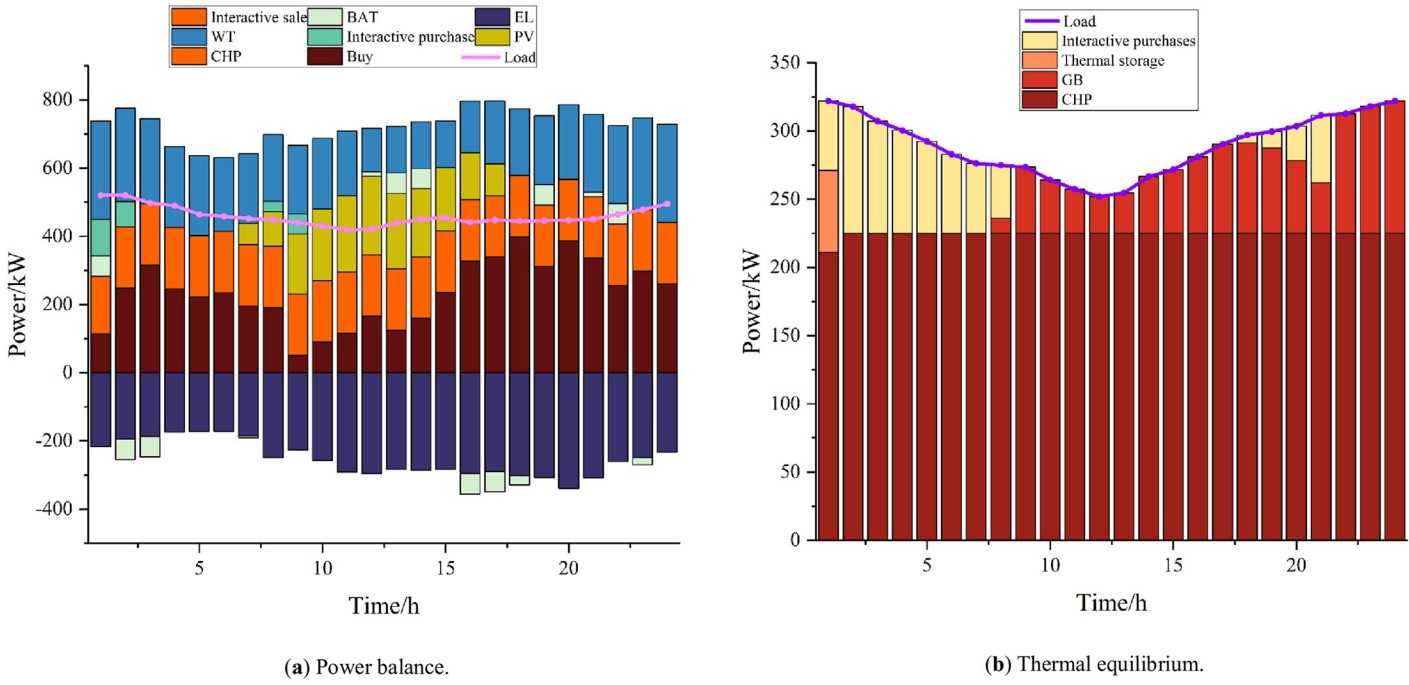

**(a)** Power balance.

**(b)** Thermal equilibrium.

**Fig 6. Industrial microgrids.**

microgrids and the system as a whole, in order to prove the effectiveness and feasibility of the strategies proposed in this paper.

As can be seen from Table 6, the cost of the joint operation of the multi-microgrid is USD 32,508.98, which is 1,147.42 lower compared to the cost of the independent operation of the multi-microgrid, a decrease of about 3.41%. Also, the carbon emissions of the multi-microgrid cluster system decreased from 13,556.24 kg to 12,590.44 kg, a decrease of about 7.12%. This indicates that renewable energy can be better utilized for power generation, and more renewable energy can be used in interactive operation, thus enabling an overall improvement in the economy and environment.

**5.1.2. Microgrid interaction analysis.** Fig 7 gives the results of electric and thermal energy interaction and scheduling between each microgrid under the cooperative operation of multi-microgrid systems. The energy interactions among microgrids are mainly concentrated in the period 21:00–7:00; 24:00–2:00 is the time period when commercial microgrids are in the off-peak state, so in addition to meeting their own power demand, they can sell power to the industrial microgrids that are still producing at night in order to gain revenue. Due to the principle of CHP's "heat to set power" and the fact that the commercial microgrids enter their off-

**Table 6. Costs of each microgrid under different modes of operation.**

| Running Costs of Different Approaches | Run independently | Joint operation |
|---|---|---|
| Resident micro-network | 13472 | 12117.4 |
| Business micro-network | 6820.3 | 7581.53 |
| Industrial micro-networks | 13364.1 | 12810.05 |
| total system | 33656.4 | 32508.98 |
| Total carbon emissions/kg | 13556.24 | 12590.44 |

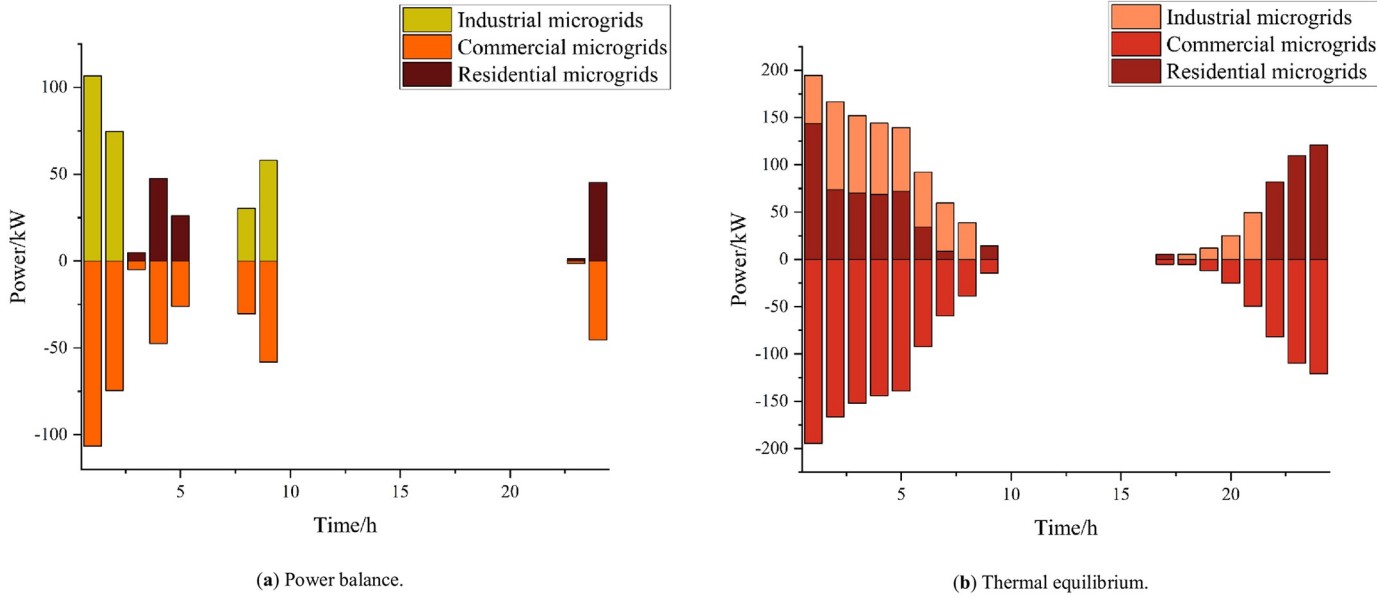

(a) Power balance.

(b) Thermal equilibrium.

**Fig 7. Microgrid interaction.**

peak period from 20:00 to 8:00, they will be in the off-peak state at this time. Due to the principle of CHP's "heat for power" and the fact that the commercial microgrids are closed from 20:00 to 8:00, these commercial microgrids sell heat to industrial microgrids and residential microgrids during this time to gain revenue. From the above analysis, it can be seen that the development of energy interaction can promote the balance of energy supply and demand, and realize the economic and flexible operation of multi-dimensional energy.

## 5.2. Analysis of different penetration rates of hydrogen-fueled vehicles

As the penetration rate of hydrogen-fueled vehicles increases, we will see an impact on the interaction of multi-microgrid systems. Here, the microgrid interactions are analyzed with consideration of different penetration rates of hydrogen-fueled vehicles, and the industrial microenergy network loads with 13% and 32% penetration rates are shown in Fig 8.

Fig 9 shows the trends in the capacity of each microgrid to power hydrogen-fueled vehicles at 13% penetration; compared to 16% penetration, here, the microgrid power is reduced. The industrial microgrid buys 75 kW of power from the commercial microgrid, but only buys power during the period of 8:00–9:00. All the power sold by the commercial microgrid is sent to the industrial microgrid, except for within the period around 20:00, and the commercial microgrid sells all its power to the industrial microenergy network. At 13% penetration, the industrial microgrid only purchases thermal energy at 20:00.

Fig 10 shows the interactions of each microgrid's capacity at a 32% penetration of hydrogen-fueled vehicles. The microgrid's electrical energy increases under these conditions compared to 16% penetration; as the loads on the campus increase, the industrial microgrid must obtain electrical energy from the commercial microgrid to meet its requirements. Compared to the 13% penetration rate, here, the industrial microgrid heat load increases. The commercial microgrid is not subjected to a large heat load at night, and during this time the industrial microgrid makes heat purchases from the commercial microgrid.

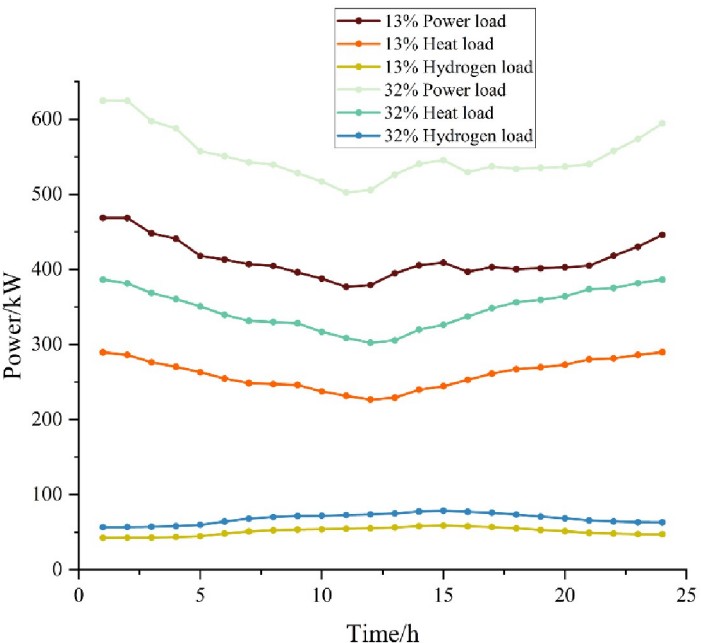

**Fig 8. Industrial microenergy network loads at different penetration rates.**

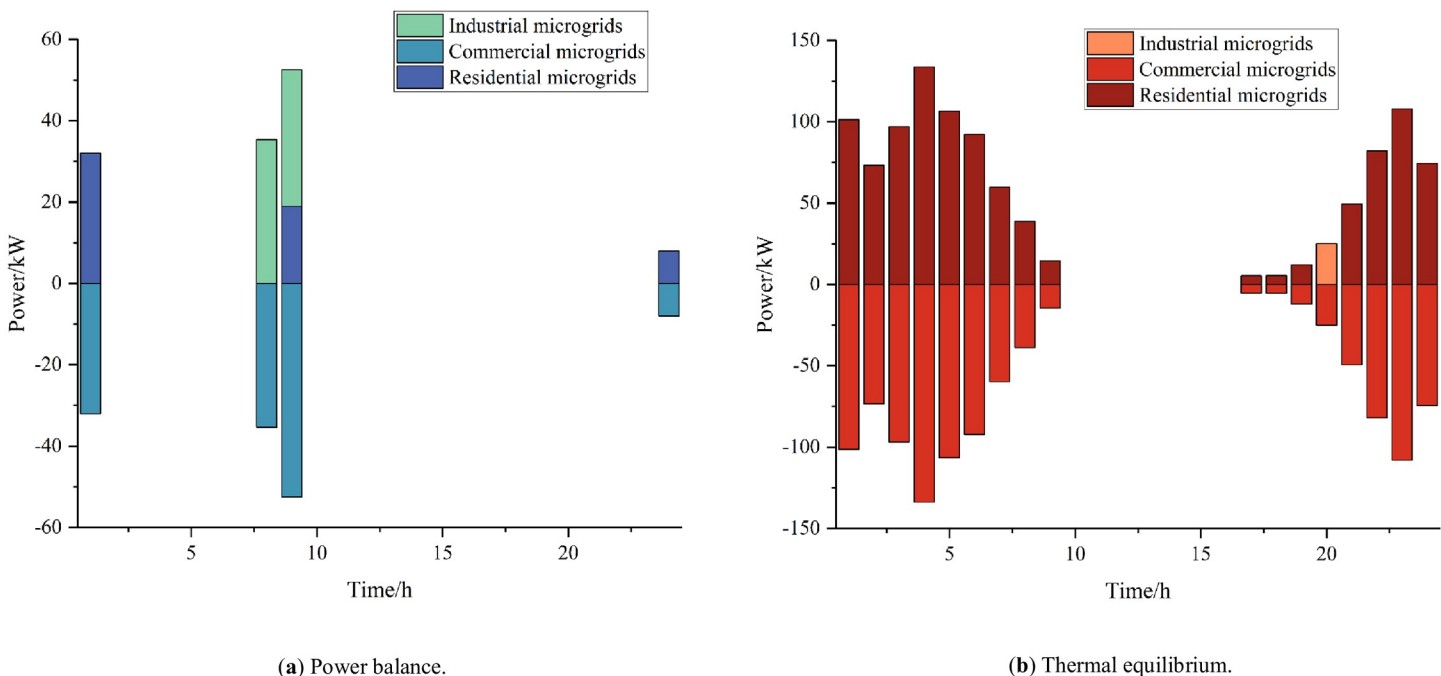

(**a**) Power balance.

(**b**) Thermal equilibrium.

**Fig 9. Microgrid interactions at 13% penetration rate.**

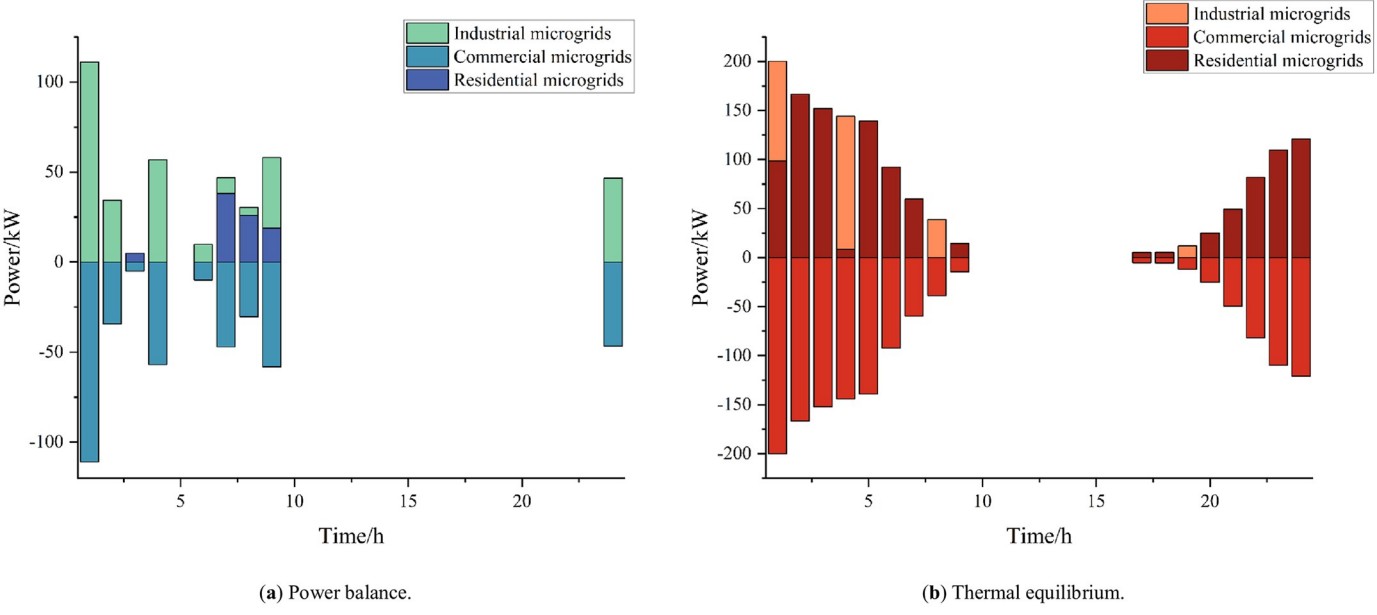

(a) Power balance.                                                    (b) Thermal equilibrium.

**Fig 10. Microgrid interactions at 32% penetration rate.**

As the penetration of hydrogen-fueled vehicles increases, the loads on industrial microenergy networks increase, leading to a gradual increase in commercial microenergy network interactions.

## 6. Conclusion

For the optimal operation of heterogeneous multi-microgrid systems containing hydrogen-fueled vehicles, this paper proposes an optimal scheduling strategy of power–heat–carbon for interconnected heterogeneous multi-microgrids containing hydrogen-fueled vehicles. Based on the results of the arithmetic example, the following conclusions are drawn:

1. The interconnected heterogeneous multi-microgrid model constructed in this paper takes into account the diversity of electric and thermal loads within the multi-microgrid system and the differences in the loads of different microgrids, and it improves the utilization efficiency of energy in the time dimension.

2. Compared with independently operated microgrids, the cooperatively operated multi-microgrid system can realize the spatial transfer of energy, which greatly improves the efficiency of new energy consumption, realizes overall improvements in the economic and environmental nature of the multi-microgrid's system operation, and downgrades the planning costs.

3. As the degree of penetration of hydrogen-fueled vehicles increases, the interaction amongst industrial microenergy networks gradually increases, and the energy interaction between commercial and industrial microenergy networks increases due to the lower load placed on the commercial microenergy network at night in order to maintain its economic operation.

The models mentioned in this paper may lead to uneven returns within the system. In the future, cooperative games within the system will be considered to maximize the benefits of

each functional area. Financial tools such as virtual bidding will be used to further improve the economic efficiency.

## Supporting information

**S1 Data.**
(XLSX)

## Author Contributions

**Conceptualization:** Wentao Huang.

**Data curation:** Xinyu Liu.

**Methodology:** Dahu Li, Zirui Shao.

**Resources:** Wentao Huang, Bohan Zhang, Jun He.

**Supervision:** Dahu Li.

**Validation:** Bohan Zhang, Jun He.

**Writing – original draft:** Dahu Li, Zirui Shao.

**Writing – review & editing:** Dahu Li, Zirui Shao, Wentao Huang.

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
