## [Decision Letter · Decision Letter 0]

26 Feb 2024

PONE-D-24-00542Optimal electricity-heat-carbon scheduling strategy for interconnected heterogeneous multi-micro-energy networks considering hydrogen fuel vehiclesPLOS ONE

Dear Dr. Shao,

Thank you for submitting your manuscript to PLOS ONE. After careful consideration, we feel that it has merit but does not fully meet PLOS ONE’s publication criteria as it currently stands. Therefore, we invite you to submit a revised version of the manuscript that addresses the points raised during the review process.

We look forward to receiving your revised manuscript.

Kind regards,

Lei Chen, Ph.D.

Academic Editor

PLOS ONE

3. Please note that your Data Availability Statement is currently missing [the repository name and/or the DOI/accession number of each dataset OR a direct link to access each database]. If your manuscript is accepted for publication, you will be asked to provide these details on a very short timeline. We therefore suggest that you provide this information now, though we will not hold up the peer review process if you are unable.

4. PLOS requires an ORCID iD for the corresponding author in Editorial Manager on papers submitted after December 6th, 2016. Please ensure that you have an ORCID iD and that it is validated in Editorial Manager. To do this, go to ‘Update my Information’ (in the upper left-hand corner of the main menu), and click on the Fetch/Validate link next to the ORCID field. This will take you to the ORCID site and allow you to create a new iD or authenticate a pre-existing iD in Editorial Manager. Please see the following video for instructions on linking an ORCID iD to your Editorial Manager account: https://www.youtube.com/watch?v=_xcclfuvtxQ.

Additional Editor Comments:

Please carefully address the comments of the reviewers to update the paper.

Reviewers' comments:

Reviewer's Responses to Questions

**Comments to the Author**

1. Is the manuscript technically sound, and do the data support the conclusions?

Reviewer #1: Yes

Reviewer #2: Yes

2. Has the statistical analysis been performed appropriately and rigorously? 

Reviewer #1: Yes

Reviewer #2: Yes

3. Have the authors made all data underlying the findings in their manuscript fully available?

Reviewer #1: Yes

Reviewer #2: Yes

4. Is the manuscript presented in an intelligible fashion and written in standard English?

Reviewer #1: Yes

Reviewer #2: No

5. Review Comments to the Author

Reviewer #1: This paper Investigated an optimal scheduling strategy of multi-energy systems considering hydrogen-fueled vehicles. It is well organized and the topic is of interest. But I still have some concerns listed below:

1) The contributions listed in the end of Section 1 should be further clarified. The specific advantages or novelties of the proposed energy scheduling framework need be highlighted.

2) The case study of analyzing the whole system interaction under different hydrogen fuel vehicle penetration rates should not be regarded as a contribution from theory or methodology perspectives.

3) The existing research works on energy scheduling strategy still need be further enriched. For instance, the robust scheduling strategy, CvaR-constrained stochastic scheduling strategy, and windfall profit-aware stochastic scheduling strategy are recommended to be mentioned.

4) The simulation setup is recommended to be further described in detail considering the specific versions of the software and solvers, which is helpful for the readers to reproduce the results.

5) The formats are recommended to be double checked and ensure consistent. The font size in some figures is too small.

6) Future research works on optimal scheduling strategy of multi-energy systems can be further recommended after the conclusion. For instance, the physical resources like flexible industrial demand resources, and the financial tools like virtual bidding, can be utilized to further increase the economic benefits.

Reviewer #2: 1- Define abbreviations at first mention. For example, MMG is defined at second mention.

2- There is no space in the beginning of the sentences.

3- Authors should use more related papers for expression literature review. Please provide one table in literature review and compare proposed approach with other works.

4- In describing the equations, please mention to the equation. For example: in equation (1), ….

5- What do you mean by " the outside world" in section "4.3.1. CHP constraints".

6- It is better to give input data, such as thermoelectric conversion efficiency, in section 5.

7- Mathematical model of multi energy system does not include any references. Please cite some relevant papers in this section. For example, you can cite, "Optimal stochastic scheduling of a multi-carrier multi-microgrid system considering storages, demand responses, and thermal comfort" published in "Sustainable Cities and Society" for energy storage devices.

8- Please use binary variables instead of 0-1 variables in section "4.4. Model solution".

9- The author claimed that the problem is linear, however, in equation 5 there is multiplication of two binary variables. ∗ = 0.

10- The manuscript needs revision for language and grammar.

6. PLOS authors have the option to publish the peer review history of their article (what does this mean?). If published, this will include your full peer review and any attached files.

Reviewer #1: No

Reviewer #2: **Yes: **Seyed Reza Seyednouri

---

## [Author Response · Author response to Decision Letter 0]

9 Mar 2024

Reviewer #1: This paper Investigated an optimal scheduling strategy of multi-energy systems considering hydrogen-fueled vehicles. It is well organized and the topic is of interest. But I still have some concerns listed below:

Comment 1: 

The contributions listed in the end of Section 1 should be further clarified. The specific advantages or novelties of the proposed energy scheduling framework need be highlighted.

Response 1:

Thank you for your attention, dear reviewer. The contributions presented in this paper have been modified to denote the uniqueness and strengths of the proposed framework. Deletions are in red and additions in yellow.

(1) Considering the characteristic endowments of residential, commercial and industrial areas, specific electric-thermal models are constructed for each type of functional area, and a framework for the operation and scheduling of heterogeneous multi-microgrid systems is designed;

(2) Considering the energy complementary characteristics of heterogeneous functional zones, formulate the electricity-heat-carbon trading strategy for multi-microgrid systems, fully utilize the differences between functional zones, and comprehensively reduce the system operating costs and carbon emissions;

(3) The impact on the whole system interaction was analyzed under different hydrogen fuel vehicle penetration rates.

(1) Considering the characteristic endowment of residential, commercial and industrial areas, construct specific power and heat models for each functional area, and design the operation and scheduling framework of heterogeneous multi-microgrid system, and construct distributed photovoltaic power generation models for residential and commercial areas, and wind power and photovoltaic power generation models for industrial areas, so as to give full play to the unique geographic advantages and characteristics of each functional area;

(2) Considering the energy complementary characteristics of heterogeneous functional zones, formulate the electricity-heat-carbon trading strategy of the multi-microgrid system, make full use of the differences between functional areas, and promote the trading between functional zones to comprehensively reduce the operating costs and carbon emissions of the system;

(3) Construct a model for hydrogen energy loads containing hydrogen-fuel vehicles and industrial areas, so as to achieve the diversified use of hydrogen energy through the consumption of renewable energy and the development of green industries.

Comment 2: 

The case study of analyzing the whole system interaction under different hydrogen fuel vehicle penetration rates should not be regarded as a contribution from theory or methodology perspectives.

Response 2:

Thank you for your suggestion, dear reviewer. Deletions are in red and additions in yellow.

(1) Considering the characteristic endowments of residential, commercial and industrial areas, specific electric-thermal models are constructed for each type of functional area, and a framework for the operation and scheduling of heterogeneous multi-microgrid systems is designed;

(2) Considering the energy complementary characteristics of heterogeneous functional zones, formulate the electricity-heat-carbon trading strategy for multi-microgrid systems, fully utilize the differences between functional zones, and comprehensively reduce the system operating costs and carbon emissions;

(3) The impact on the whole system interaction was analyzed under different hydrogen fuel vehicle penetration rates.

(1) Considering the characteristic endowment of residential, commercial and industrial areas, construct specific power and heat models for each functional area, and design the operation and scheduling framework of heterogeneous multi-microgrid system, and construct distributed photovoltaic power generation models for residential and commercial areas, and wind power and photovoltaic power generation models for industrial areas, so as to give full play to the unique geographic advantages and characteristics of each functional area;

(2) Considering the energy complementary characteristics of heterogeneous functional zones, formulate the electricity-heat-carbon trading strategy of the multi-microgrid system, make full use of the differences between functional areas, and promote the trading between functional zones to comprehensively reduce the operating costs and carbon emissions of the system;

(3) Construct a model for hydrogen energy loads containing hydrogen-fuel vehicles and industrial areas, so as to achieve the diversified use of hydrogen energy through the consumption of renewable energy and the development of green industries.

Comment 3: 

The existing research works on energy scheduling strategy still need be further enriched. For instance, the robust scheduling strategy, CvaR-constrained stochastic scheduling strategy, and windfall profit-aware stochastic scheduling strategy are recommended to be mentioned.

Response 3:

Thank you for your attention and suggestion, dear reviewer. Deletions are in red and additions in yellow.

Literature [7] considered a multi-microgrid optimization model for distributed robust optimization to solve the uncertainty optimization problem for renewable energy, loads and electric vehicles. Literature [8] proposed a two-stage trading model for microgrid clusters based on the price trading mechanism and conditional value-at-risk (CVaR) theory, which introduces the CVaR theory to quantify the potential risk of trading defaults and reduces the operating costs of microgrid clusters. Literature [9] proposes a new integrated concept of risk-seeking/averse preferences and establishes an unexpected profit-aware stochastic dispatch model for industrial virtual power plants (IVPP) based on this type of risk preferences, which enables joint management of potentially high profits and extreme losses. 

Table 1 compares the advantages of the proposed approach with the existing literature on multi-microenergy grid systems. Table 1 can be seen in the response letter.

Comment 4: 

The simulation setup is recommended to be further described in detail considering the specific versions of the software and solvers, which is helpful for the readers to reproduce the results.

Response 4:

Thank you for your attention and suggestion, dear reviewer. The version number of the simulation software used has been indicated in the paper. Deletions are in red and additions in yellow.

After the above analysis, it can be seen that the objective function and constraints in this paper are all linear, and none of them involves the multiplication term of binary variables and continuous variables, and the commercial solver Cplex integrates the advantages of the optimization algorithms such as branch-and-bound method and cut-plane method, and it has the ability of solving mixed-integer linear programming problems quickly. therefore, we use Yalmip+Cplex in MATLAB to solve the model. Therefore, CPLEX 12.6.0 was invoked in MATLAB R2016b to solve the model.

Comment 5: 

The formats are recommended to be double checked and ensure consistent. The font size in some figures is too small.

Response 5:

Thank you for your attention and suggestion, dear reviewer. Revisions have been made.

Comment 6: 

Future research works on optimal scheduling strategy of multi-energy systems can be further recommended after the conclusion. For instance, the physical resources like flexible industrial demand resources, and the financial tools like virtual bidding, can be utilized to further increase the economic benefits.

Response 6:

Thank you for your attention and suggestion, dear reviewer. An outlook for future research work on optimal scheduling strategies for multi-energy systems has been added at the end of the article. The additions are shown in yellow.

The models mentioned in this paper may lead to uneven returns within the system. In the future, cooperative games within the system will be considered to maximize the benefits of each functional area of the system. And financial tools such as virtual bidding will be used to further improve the economic efficiency.

 

Reviewer 2:

Comment 1: 

Define abbreviations at first mention. For example, MMG is defined at second mention.

Response 1:

Thank you for your attention and suggestion, dear reviewer. Abbreviations have been defined in the first mention.

Comment 2: 

There is no space in the beginning of the sentences.

Response 2:

Thank you for your attention and suggestion, dear reviewer. The spaces have been removed.

Comment 3: 

Authors should use more related papers for expression literature review. Please provide one table in literature review and compare proposed approach with other works.

Response 3:

Thank you for your attention and suggestion, dear reviewer. 

Deletions are in red and additions in yellow.

Literature [7] considered a multi-microgrid optimization model for distributed robust optimization to solve the uncertainty optimization problem for renewable energy, loads and electric vehicles. Literature [8] proposed a two-stage trading model for microgrid clusters based on the price trading mechanism and conditional value-at-risk (CVaR) theory, which introduces the CVaR theory to quantify the potential risk of trading defaults and reduces the operating costs of microgrid clusters. Literature [9] proposes a new integrated concept of risk-seeking/averse preferences and establishes an unexpected profit-aware stochastic dispatch model for industrial virtual power plants (IVPP) based on this type of risk preferences, which enables joint management of potentially high profits and extreme losses. 

Table 1 compares the advantages of the proposed approach with the existing literature on multi-micro-energy grid systems. Table 1 can be seen in the response letter

Comment 4: 

In describing the equations, please mention to the equation. For example: in equation (1), ….

Response 4:

Thank you for your attention and suggestion, dear reviewer. Changes have been made.

Comment 5: 

What do you mean by " the outside world" in section "4.3.1. CHP constraints".

Response 5:

Thank you for your attention and suggestion, dear reviewer. This was an error and has been changed. Deletions are in red and additions in yellow.

The CHP unit mainly contains a gas turbine and a boiler that provides electricity and heat to the outside world and is mathematically modeled:

The CHP unit consists mainly of a gas turbine and boiler that provides power and heat to the outside, and is mathematically modelled as:

Comment 6: 

It is better to give input data, such as thermoelectric conversion efficiency, in section 5.

Response 6:

Thank you for your attention and suggestion, dear reviewer. The efficiency parameters appearing in the model in the text have been adjusted to section 5.1. Table 2 can be seen in the response letter

Comment 7: 

Mathematical model of multi energy system does not include any references. Please cite some relevant papers in this section. For example, you can cite, "Optimal stochastic scheduling of a multi-carrier multi-microgrid system considering storages, demand responses, and thermal comfort" published in "Sustainable Cities and Society" for energy storage devices.

Response 7:

Thank you for your attention and suggestion, dear reviewer. Relevant quotes have been added to the mathematical model of the article. Added in section 4.3.4 "Optimal stochastic scheduling of a multi-carrier multi-microgrid system considering storages, demand responses, and thermal comfort" published in "Sustainable Cities and Society"

[21]Liu, Ming, Shan Wang, and Junjie Yan. "Operation scheduling of a coal-fired CHP station integrated with power-to-heat devices with detail CHP unit models by particle swarm optimisation algorithm." Energy 214 (2021): 119022.

[22]li, yang, et al. "Improving operational flexibility of integrated energy system with uncertain renewable generations considering thermal inertia of buildings." Energy Conversion and Management 207 (2020): 112526. 23.

[23] Seyednouri, S. R., et al. "Optimal stochastic scheduling of a multi-carrier multi-microgrid system considering storages, demand responses, and thermal comfort." Sustainable Cities and Society 99 (2023): 104943.

Comment 8: 

Please use binary variables instead of 0-1 variables in section "4.4. Model solution".

Response 8:

Thank you for your attention and suggestion, dear reviewer. Thank you for your attention and suggestion, dear reviewer. The version number of the simulation software used has been indicated in the paper. Deletions are in red and additions in yellow.

After the above analysis, it can be seen that the objective function and constraints in this paper are all linear, and none of them involves the multiplication term of binary variables and continuous variables, and the commercial solver Cplex integrates the advantages of the optimization algorithms such as branch-and-bound method and cut-plane method, and it has the ability of solving mixed-integer linear programming problems quickly. therefore, we use Yalmip+Cplex in MATLAB to solve the model.Therefore, CPLEX 12.6.0 was invoked in MATLAB R2016b to solve the model.

Comment 9: The author claimed that the problem is linear, however, in equation 5 there is multiplication of two binary variables. ∗ = 0.

Response 9:

Thank you for your attention and suggestion, dear reviewer. A modification has been made in Equation 5, where the multiplication sign has been changed to a plus sign.

Comment 10: The manuscript needs revision for language and grammar.

Response 10:

Thank you for your attention and suggestion, dear reviewer. The language of the article has been revised and enhanced. The image can be seen in the response letter

---

## [Decision Letter · Decision Letter 1]

14 Mar 2024

Optimal power–heat–carbon scheduling strategy for interconnected heterogeneous multi-microgrid considering hydrogen fuel cell vehicles

PONE-D-24-00542R1

Dear Dr. Shao,

We’re pleased to inform you that your manuscript has been judged scientifically suitable for publication and will be formally accepted for publication once it meets all outstanding technical requirements.

Kind regards,

Lei Chen, Ph.D.

Academic Editor

PLOS ONE

Additional Editor Comments:

The authors have well addressed the comments of the reviewers, and the revised paper can be accepted for publication.

Reviewers' comments:

Reviewer's Responses to Questions

**Comments to the Author**

1. If the authors have adequately addressed your comments raised in a previous round of review and you feel that this manuscript is now acceptable for publication, you may indicate that here to bypass the “Comments to the Author” section, enter your conflict of interest statement in the “Confidential to Editor” section, and submit your "Accept" recommendation.

Reviewer #1: All comments have been addressed

Reviewer #2: All comments have been addressed

2. Is the manuscript technically sound, and do the data support the conclusions?

Reviewer #1: Yes

Reviewer #2: Yes

3. Has the statistical analysis been performed appropriately and rigorously? 

Reviewer #1: N/A

Reviewer #2: I Don't Know

4. Have the authors made all data underlying the findings in their manuscript fully available?

Reviewer #1: Yes

Reviewer #2: Yes

5. Is the manuscript presented in an intelligible fashion and written in standard English?

Reviewer #1: Yes

Reviewer #2: Yes

6. Review Comments to the Author

Reviewer #1: The quality and significance of this manuscript has been improved considering my previous comments. No additional comments.

Reviewer #2: (No Response)

7. PLOS authors have the option to publish the peer review history of their article (what does this mean?). If published, this will include your full peer review and any attached files.

Reviewer #1: No

Reviewer #2: **Yes: **Seyed Reza Seyednouri

---

## [Editor Report · Acceptance letter]

21 Mar 2024

PONE-D-24-00542R1 

PLOS ONE

Dear Dr. Shao, 

I'm pleased to inform you that your manuscript has been deemed suitable for publication in PLOS ONE. Congratulations! Your manuscript is now being handed over to our production team.

Kind regards, 

on behalf of

Professor Lei Chen 

Academic Editor

PLOS ONE